# A novel network control model for identifying personalized driver genes in cancer

Wei-Feng Guo[1], Shao-Wu Zhang[1]*, Tao Zeng[2,3], Yan Li[1], Jianxi Gao[4,5]*, Luonan Chen[1,2,3,6]*

**1** Key Laboratory of Information Fusion Technology of Ministry of Education, School of Automation, Northwestern Polytechnical University, Xian, China, **2** Key Laboratory of Systems Biology, Center for Excellence in Molecular Cell Science, Shanghai Institutes for Biological Science, Chinese Academy Science, Shanghai, China, **3** Shanghai Research Center for Brain Science and Brain-Inspired Intelligence, Shanghai, China, **4** Department of Computer Science, Rensselaer Polytechnic Institute, Troy, New York, United States of America, **5** Network Science and Technology Center, Rensselaer Polytechnic Institute, Troy, New York, United States of America, **6** Center for Excellence in Animal Evolution and Genetics, Chinese Academy of Sciences, Kunming, China

* zhangsw@nwpu.edu.cn (S-WZ); gaoj8@rpi.edu (JG); lnchen@sibs.ac.cn (LC)

## Abstract

Although existing computational models have identified many common driver genes, it remains challenging to identify the personalized driver genes by using samples of an individual patient. Recently, the methods of exploiting the structure-based control principles of complex networks provide new clues for identifying minimum number of driver nodes to drive the state transition of large-scale complex networks from an initial state to the desired state. However, the structure-based network control methods cannot be directly applied to identify the personalized driver genes due to the unknown network dynamics of the personalized system. Here we proposed the personalized network control model (PNC) to identify the personalized driver genes by employing the structure-based network control principle on genetic data of individual patients. In PNC model, we firstly presented a paired single sample network construction method to construct the personalized state transition network for capturing the phenotype transitions between healthy and disease states. Then, we designed a novel structure-based network control method from the Feedback Vertex Sets-based control perspective to identify the personalized driver genes. The wide experimental results on 13 cancer datasets from The Cancer Genome Atlas firstly showed that PNC model outperforms current state-of-the-art methods, in terms of F-measures for identifying cancer driver genes enriched in the gold-standard cancer driver gene lists. Furthermore, these results showed that personalized driver genes can be explored by their network characteristics even when they are hidden factors in transcription and mutation profiles. Our PNC gives novel insights and useful tools into understanding the tumor heterogeneity in cancer. The PNC package and data resources used in this work can be freely downloaded from https://github.com/NWPU-903PR/PNC.

**Data Availability Statement:** All relevant data are within the manuscript and its Supporting Information files.

**Funding:** This paper was supported by the National Natural Science Foundation of China (61873202,

61473232, 91430111, 31930022, 31771476, 81471047 and 11871456) and National Key R&D Program (2017YFA0505500), Shanghai Municipal Science and Technology Major Project (No. 2017SHZDZX01), National Key R&D Program (Special Project on Precision Medicine) (2016YFC0903400), and Natural Science Foundation of Shanghai (17ZR1446100). All the above funders played roles in the study design, data collection and analysis and preparation of the manuscript

**Competing interests:** The authors have declared that no competing interests exist.

## Author summary

Notably there may be unique personalized driver genes for an individual patient in cancer. Identifying personalized driver genes that lead to particular cancer initiation and progression of individual patient is one of the biggest challenges in precision medicine. However, most methods for cancer driver genes identification have focused mainly on the cohort information rather than on individual information and fail to identify personalized driver genes. We here proposed personalized network control model (PNC) to identify personalized driver genes by applying the structure based network control principle on personalized data of individual patients. By considering the progression from the healthy state to the disease state as the network control problem, our PNC aims to detect a small number of personalized driver genes that are altered in response to input signals for triggering the state transition in individual patients on expression level. The impetus behind PNC contains two main respects. One is to design a paired single sample network construction method (namely Paired-SSN) for constructing personalized state transition networks to capture the phenotypic transitions between normal and disease attractors. The other one is to develop a novel structure based network control method (namely NCUA) on personalized state transition networks for identifying personalized driver genes which can drive individual patient system state from healthy state to disease state through oncogene activations. Each part of the proposed method has been deeply examined to be efficient. Compared with other existing models, our PNC shows a higher performance in terms of F-measures of the cancer driver genes in the well-known Cancer Census Genes (CCG) and Network of Cancer Genes (NCG). The wide experimental results on multiple cancer datasets highlight that sample specific network theory and structure based network control theory can contribute to identifying personalized driver genes in cancer.

## Introduction

Cancer is a heterogeneous disease that is driven by oncogene activations such as genetic mutation, gene amplification, chromosomal rearrangement, and transposable elements[1–3]. During tumor progression, the majority of detected altered genes are passengers that do not contribute to the oncogenic process, but a small fraction of genomic and transcriptomic altered genes are known as driver genes that modify transcriptional programs and therefore drive and sustain tumor progression from healthy state to disease state. Many bioinformatics tools for driver gene identification with multi-dimensional genomic data have been developed recently. To our best understanding, we categorized these approaches into two groups according to their significant features. (i) The driver gene identification methods in large cohorts, such as mutation frequency-based methods and machine learning-based methods. The mutation frequency-based methods mainly identify the drive genes by finding significantly mutated genes whose mutation rates are significantly higher than the background mutation rate [4, 5]. Due to tumor heterogeneity, constructing a reliable background mutation model is difficult, which limits the performance of frequency-based methods. On the other hand, the machine learning-based methods can be developed for any specific tasks depending on the available training data designed as pathogenic or neutral, but they have a few applications due to the probable incompleteness of their referred databases [6, 7]. (ii) The driver gene identification methods for individual patients. These methods such as SCS [8], DawnRank [9] assume that cancer is a complex disease with many changes altered at the network level and identify personalized driver genes by integrating personalized genetic data of an individual patient and the

reference gene (or protein) interaction data. Although these methods have been successfully used for prioritizing cancer driver genes, the ultimate goal of discovering a complete catalog of driver genes indeed associated with individual patients is far from being achieved. Therefore, new and efficient computational models and methods are urgently needed to identify personalized driver genes for understanding tumor heterogeneity in cancer.

Recently, with the development of network science, structure-based network control methods of exploiting structural controllability of complex networks have offered powerful mathematical frameworks to understand diverse biological systems at a network level [10]. So far, the structure-based network control methods which aim to find a minimum set of driver nodes for steering the states of large-scale networks to the desired states can be mainly divided into two categories according to the network styles. One focuses on the directed (or non-symmetrical) networks, and the other focuses on the undirected (or symmetrical) networks. For directed networks, some linear or local nonlinear structural control tools based on Maximum Matching Sets (MMS) are developed to identify the minimum number of driver nodes that need to be injected by external signals to achieve the desired control objectives [11–13]. Although those MMS based structural network control tools have many applications in biomolecular systems [8, 14, 15], those tools may only give an incomplete view of the network control properties for a system with nonlinear dynamics. Recently, the method of Feedback Vertex Set (FVS)-based control (FC) [16, 17] is proposed to control the large-scale networks in a reliable and nonlinear manner, where the network structure is prior known and the functional form of the governing equations is not specified, but it must satisfy some properties. Within such a scheme, a Directed FVS-based control method (DFVS) is applied to control the directed networks by identifying the source nodes and FVS nodes as the driver nodes [18]. The above approaches only focus on the mainly linear or nonlinear dynamics of the directed networks. There are few approaches to investigate the structural controllability of the undirected networks. For example, the method of Minimum Dominating Sets (MDS) [19] investigates the structural controllability of the undirected networks. Since it works with the strong assumption that the controllers can control its outgoing links independently, MDS requires higher costs in many kinds of networks which may underestimate the structural controllability of undirected networks. Therefore, efficient methods need to be developed for studying the structural controllability of the undirected networks.

Considering above facts together, structure-based network control methods can help the personalized driver genes identification in cancer propagation, but they cannot be directly applied to the personalized driver genes identification of individual patients. This is primarily due to a gap between the structure based network control methods and the applications in the personalized patient system. The rate-limiting step of structure based network control methods is that efficient methods are not available to obtain personalized state transition networks that capture the phenotypic transitions between healthy and disease states. We addressed the limitation by introducing a novel personalized network control model (PNC), which sheds light on the cancer transition network between a disease state and a healthy state by employing the network control principle on the personalized genetic data of individual patients. PNC consists of the following two steps: (i) construct a personalized state transition network for characterizing the state transition of each individual during cancer progression and (ii) design structure control method on the personalized state transition network. That is, we firstly developed a paired single sample network construction method (namely, Paired-SSN) to construct the personalized state transition network for capturing the phenotype transitions between normal state and tumor state through the integration of the expression data and gene/protein interaction network. The personalized state transition network is a graph in which nodes represent genes, and edges denote the significant co-expression difference between normal state

and tumor state in the gene interaction network. Then, on the personalized state transition networks, we designed a novel structure based network control method (namely NCUA) according to the FC theory to drive a complex undirected network with nonlinear dynamics from initial attractor to desired attractor through perturbation to a feasible subset of driver nodes.

By our PNC method, the wide experimental results from 13 kinds of cancer datasets in The Cancer Genome Atlas (TCGA) demonstrate that:

i.  Considering the Cancer Census Genes (CCG) [20] and Network of Cancer Genes (NCG) [21] as the gold-standard of cancer driver genes, PNC performs better in terms of the F-measures for identifying the cancer driver genes on multiple cancer datasets than nine cancer driver genes focus methods (i.e., DriverML [7], SCS [8], DawnRank [9], MutSigCV [22], ActiveDriver [23], DriverNet [24], OncoDriveFM [25], SSN [26] and Mutation frequency based method) and four traditional personalized driver genes focus methods (i.e., three Differential expression genes identification methods and Hub genes selection method).

ii.  Each part of the PNC model (Paired-SSN and NCUA) has been examined to be efficient. The F-measures of NCUA method for identifying the personalized driver genes enriched in CCG and NCG genes would be higher when Paired-SSN rather than LIONESS and SSN is used. When Paired-SSN is used for constructing the personalized state transition network, NCUA can more effectively improve the performance than other network control methods, in terms of the F-measures for identifying the personalized driver genes that are enriched in the CCG and NCG genes.

iii.  The effect of different gene interaction networks on the performance of PNC is also evaluated. The results on multiple gene interaction networks indicate that a proper reference gene interaction network is a key factor for PNC on identifying cancer driver genes and more complete and higher quality gene interaction information would improve PNC's prediction power.

iv.  To quantify the effect of network structure on personalized controllability, we used the connected component (CC) of each personalized state transition network to determine the network parameters of individual patient system and gave the relationship between personalized controllability and network structure on different cancer datasets. The results on multiple cancer data sets indicate that the more heterogeneous the degree distribution of constructed personalized state transition network is, the easier it is to control the entire system.

v.  By analyzing the statistic information of individual specific sub-networks of *TP53*, we found that PNC is able to explore the candidate driver genes with low differential expression fold change and low mutation frequency by their network and structural characteristics. Therefore, our PNC model provides a new powerful tool for identifying the personalized driver genes of individual patients and gives novel insights for understanding tumor heterogeneity in cancer.

## Results

### An overview of PNC model

Cancer can be perceived as a dysfunction of molecular networks that regulate molecular communications and cellular processes. To understand cancer progression, we need to understand the dynamics of these networks with respect to control theory [27]. By considering the gene

expression profiles in normal and tumor samples as the respective state for a given patient and applying adequate knowledge of the network structure, structure based network control tools aim to detect a small number of genes that are altered in response to input signals and trigger a state transition in individual patients. The application of structure based network control methods for personalized driver genes identification requires two key steps, i.e., (i) to construct personalized state transition networks which are involved in the state transition during disease development for each patient and (ii) to design the optimal network control methods based on the structure of the personalized state transition network. As a result, PNC model consists of two main parts (Fig 1). The first part is that a new Paired-SSN method rather than an original SSN [26] was developed to obtain the personalized state transition networks by using matched expression information of normal and tumor samples of an individual patient and gene interaction network. For an individual patient, we obtained co-expression p-value of the interaction edges for normal sample and tumor sample respectively by using SSN. Based on the co-expression p-value of the interaction edges for normal sample and tumor sample, the personalized state transition network was constructed. In the personalized state transition network, the edge between gene $i$ and gene $j$ will exist if co-expression p-value of the interaction edge is less than (greater than) 0.05 for the tumor sample but greater than (less than) 0.05 for the normal sample. Therefore a personalized state transition network represents which gene pairs are significant difference between normal sample and tumor sample of an individual patient.

And the second part is to design a novel Nonlinear structural Control Undirected network based Algorithm (NCUA) for identifying the personalized driver genes. For NCUA method, we assumed that each edge in an undirected network is bi-directional (feedback loop) and constructed a new bipartite graph from the original undirected network. In bipartite graph, the nodes of top side are the nodes of original graph and the nodes of the bottom side are the edges of the original graph. To identify the minimum driver nodes, we adopted an equivalent optimization procedure for obtaining the minimum dominating nodes in the up side to cover the bottom side nodes in the bipartite graph. Of note, we used a gene interaction network from literature [9] by integrating multiple types of datasets, including Mutual Exclusivity Modules (MEMo) in cancer [28, 29], Reactome [30], NCI-Nature Curated PID [31] and Kyoto Encyclopedia of Genes and Genomes [32]. Such gene interaction network consists of 11,648 genes and 211,794 interactions. Those interactions consist of gene-interaction from multiple high-quality sources such as protein interactions, gene co-expressions, TF-target interactions, protein domain interactions, text-mined interactions, etc. The integrated networks with 211,794 interactions are listed in **S1 File**.

## Biological intuition of PNC model

The biological system in an individul patient is generally a nonlinear dynamical system. From the dynamics point of view, gene expressions are variables of such a system and may be different if measured at different time points for the same patient. In contrast, it is transcriptional networks that thus result in the measured gene expression pattern and determine the state transition of an individual patient in caner development. The state transition networks or transcriptional networks of an individul patient can more reliably characterize the biological system of the individual patient. The biological interpretation of personalized state transition network represents which gene pairs are involved in the disease development for each patient.

Thus it is important to reconstruct the personalized state transition networks using the personalized genetic data (e.g., expression profiles). Currently most of the studies for exploiting gene regulation, such as the Single Sample Network (SSN) [26] and Linear Interpolation to Obtain Network Estimates for Single Samples(LIONESS) [33], can describe the dynamic gene

**(a) Motivation of PNC**

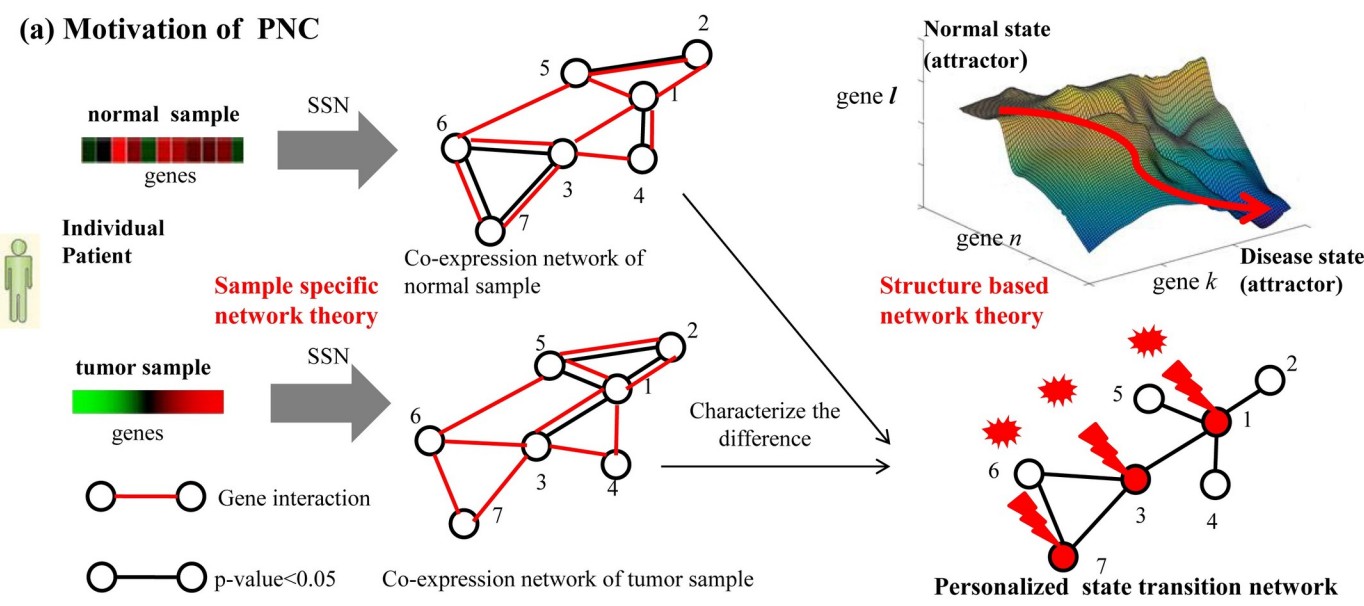

**(b) Framework of PNC**

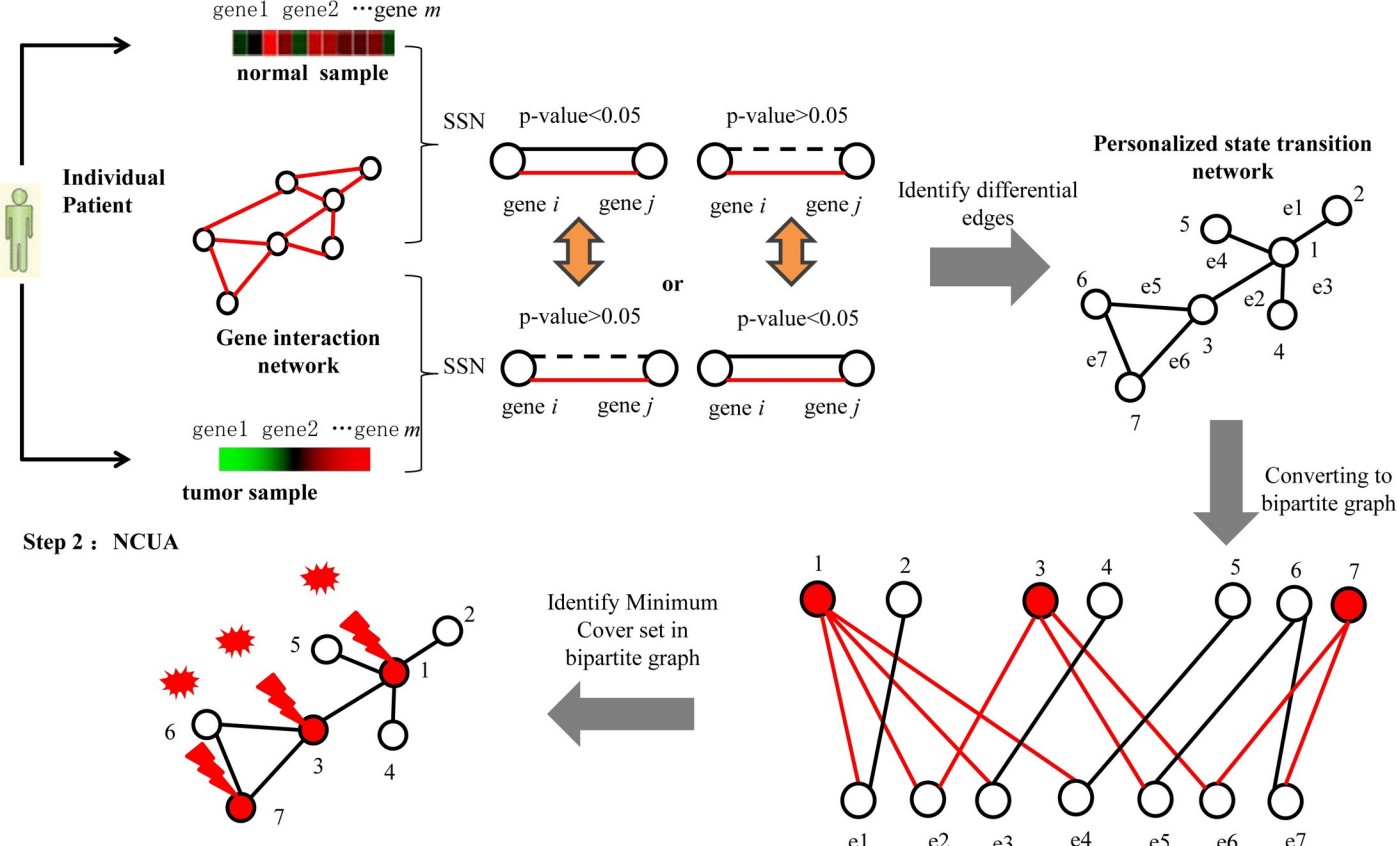

**Fig 1. Overview of personalized network control model (PNC) for identifying personalized driver genes.** (a) The motivation of PNC. By integrating sample specific network theory and structure based network control theory, a small number of personalized driver genes that are altered in response to oncogene activations is detected for triggering the state transition of an individual patient from the healthy state to the disease state at expression level. The main consideration of PNC is to design methods for i) constructing personalized state transition networks to capture the phenotypic transitions between healthy and disease states / attractors, and ii) developing network control method on personalized state transition networks where we identify personalized driver genes for driving individual system in cancer from normal attractor to disease attractor through oncogene activation. **(b)** The main workflow of PNC. Our PNC consists of two main parts. One is a paired single sample network construction method (Paired-SSN), which we used to construct personalized state transition networks. In these networks, edges denote the significant difference of gene interactions between normal sample and tumor sample of an individual patient. For the second part of PNC, we designed a novel nonlinear structural control method (namely, NCUA) for identifying personalized driver genes. This was done to ensure that the state in the personalized state transition network would asymptotically be changed from normal state to disease state through oncogene activations.

regulation for an individual patient. But they ignore normal sample information of an individual patient and consequently and they yield many false-positive calls for constructing personalized state transition network. Therefore to construct the personalized state transition network for each patient, we designed Paired-SSN method based on sample specific network theory (SSN) [26] for characterizing significant difference of gene pairs between normal sample and tumor sample of an individual patient. In Fig 2, we gave some examples how to apply Paired-SSN for constructing the personalized state transition network based on the expression data of two cancer patients (TCGA.EJ.7781).Then based structure based network control theory, we developed NCUA method to identify driver genes for driving the whole personalized state transition network state of an individual patient from the normal attractor to disease attractor. From the structure based network control respective, the personalized driver genes on the personalized state transition network of an individual patient are the altered nodes/genes in response to input signals which can trigger the state transition of the whole network. The input signals may be oncogene activation signals such as a genetic mutation, gene amplification, a chromosomal rearrangement, or transposable elements. The "controllers" in PNC model for identifying personalized driver genes are the genetic or environment factors which produce the oncogene activation signals. Therefore the biological interpretation of the PNC model is to identify personalized driver genes which can trigger the state transition of the individual system from normal attractor to disease attractor in response to oncogene activation signals.

## Performance comparisons of PNC with current state-of-the-art methods for identifying cancer driver genes

To identify personalized driver genes, we collected paired samples (a normal sample and a tumor sample) from each individual. Here we used cancer datasets which contained enough normal-disease paired samples (>20 paired samples) in TCGA for the case study. By searching TCGA, 13 cancer datasets met the requirements, i.e., the datasets for breast invasive carcinoma (BRCA), colon adenocarcinoma (COAD), kidney chromophobe (KICH) and kidney renal clear cell carcinoma (KIRC), kidney renal papillary cell carcinoma (KIRP), liver hepatocellular carcinoma (LIHC), lung adenocarcinoma (LUAD), lung squamous cell carcinoma (LUSC), stomach adenocarcinoma (STAD), uterine corpus endometrial carcinoma(UCEC), Head and Neck Squamous Cell Carcinoma (HNSC), Prostate Adenocarcinoma (PRAD) and Thyroid Papillary Carcinoma (THCA). For more information, see S1 Table of **S2 File**. On the 13 cancer datasets, the identified cancer driver genes annotated in the CCG and NCG were adopted to compute the F-measure scores (Methods) for assessing the performance of different methods. In total, we collected 616 cancer census genes and 711 known cancer genes from CCG and NCG gene lists (**S3 File**).

In the method comparisons, PNC selects the cancer driver genes which have high frequency (>0.8) among all the patients on each cancer data sets. From Fig 3, we found that the F-

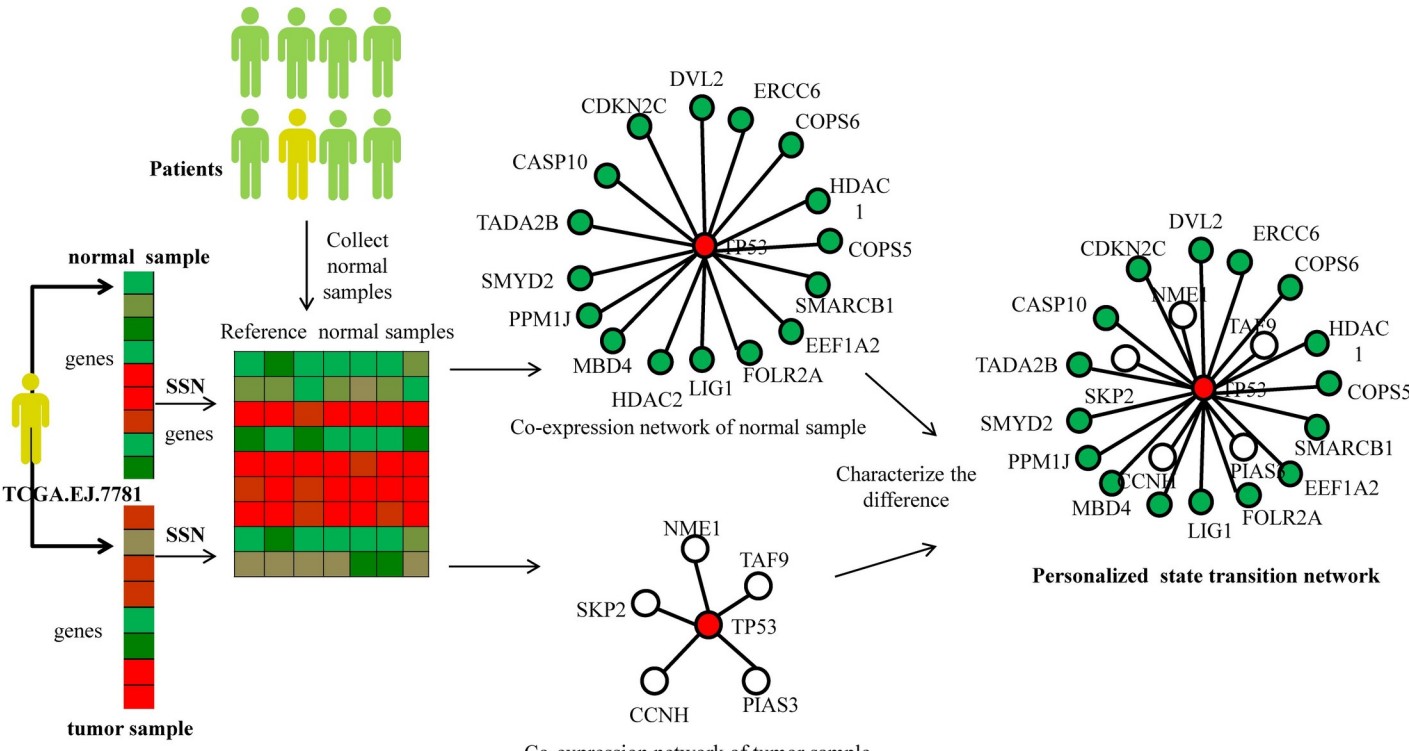

**Fig 2. Overview of Paired-SSN for constructing personalized state transition networks.** For a given cancer patient, TCGA.EJ.7781 in Prostate Adenocarcinoma (PRAD) cancer data, we chose the expression data of all normal samples in PRAD as the reference data and respectively constructed the co-expression network of tumor sample (white color) and normal sample (green color) with the reference data by using SSN method. Then the personalized state transition network was constructed where the nodes represent the genes and edges denote the significant difference of gene interactions between normal sample and tumor sample of an individual patient in the disease development. Here we showed the individual specific sub-networks related with driver gene TP53 which contain its first-order neighboring genes as an example in this Fig.

measures of PNC would be higher than 9 methods on gene mutation data (i.e., DriverML [7], SCS [8], DawnRank [9], MutSigCV [22], ActiveDriver [23], DriverNet [24], OncoDriveFM [25], SSN [26] and Mutation frequency method) and 4 methods on gene expression data (i.e., DEG-FoldChange, DEG-p-value, DEG-FDR and Hub genes selection method) on 13 cancer datasets. We showed the predicted driver genes list of different methods in **S4 File**. We obtained the cancer driver genes of DriverML [7], SCS [8], and SSN [26] from their provided driver genes list. We obtained the driver genes in DawnRank [9], MutSigCV[22], ActiveDriver [23], DriverNet[24], OncodriverFM[25], and from the DriverDBv2 database [34]. Besides, the DEG-FoldChange selects the personalized driver genes by calculating the fold-change between the normal sample and tumor sample ($|\log2(\text{fold-change})| > 1$). DEG-p-value and DEG-FDR select the personalized driver genes by respectively calculating the p-value and FDR ($<0.05$) between a cancer tumor sample and a group of control samples. Hub genes selection method regards hub genes in the constructed network as cancer driver genes (Methods). All the above methods ran the same TCGA datasets according to their manuals.

## The influence of different single sample network construction methods

In addition, to demonstrate the advantage of Paired-SSN over other methods for constructing personalized state transition networks, we also evaluated the performance of LIONESS and

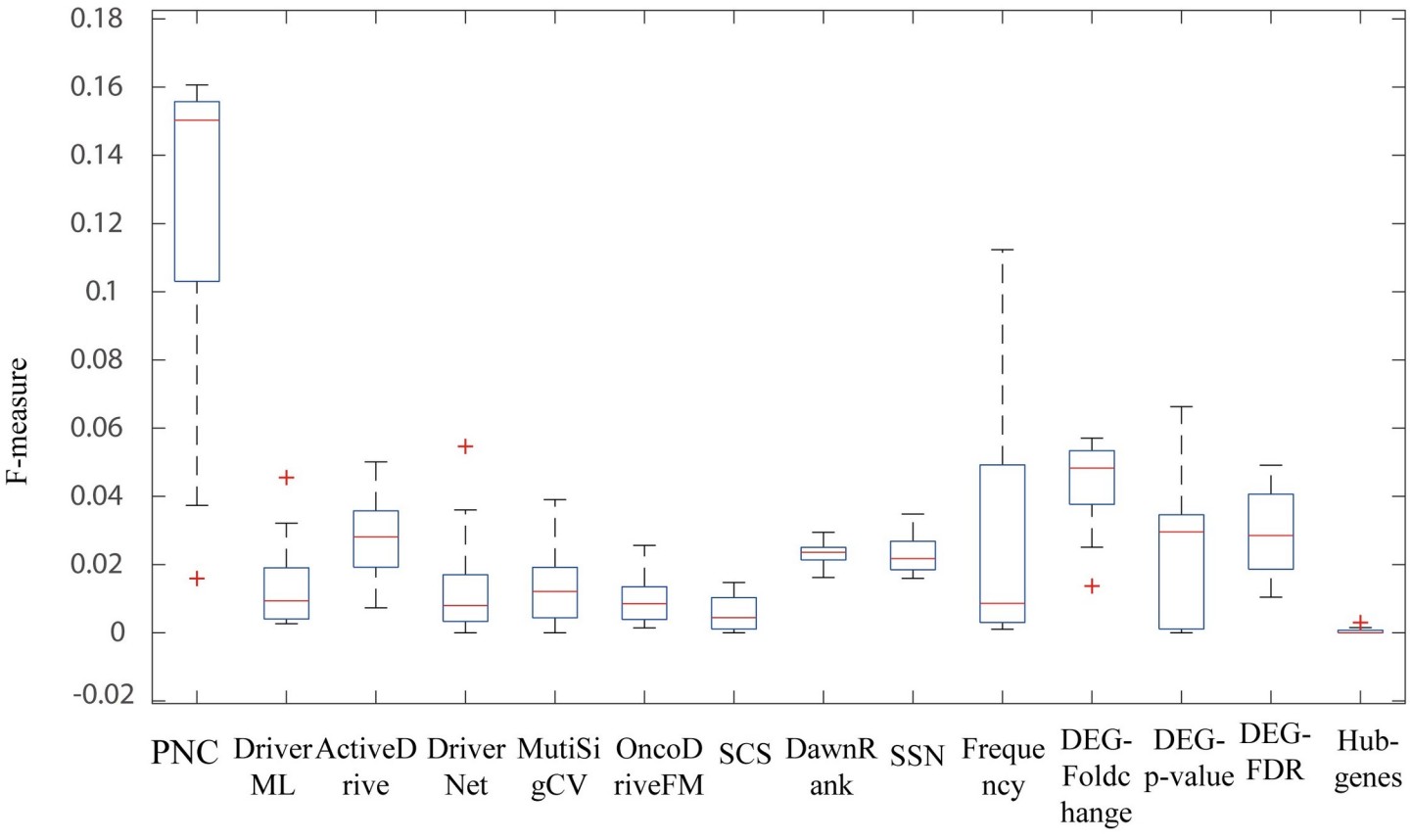

**Fig 3. The significant enrichment F-scores of PNC and other methods for identifying cancer driver genes.**

SSN on the same cancer datasets. Different from Paired-SSN, SSN considers the tumor sample network directly as the personalized state transition network. The tumor sample network is constructed by quantifying the differential network between the tumor sample and a group of normal samples. Meanwhile, LIONESS (Linear Interpolation to Obtain Network Estimates for Single Samples) [33] constructs the personalized state transition networks by calculating the edge statistical significance between all tumor samples and the tumor samples without a given single sample (Methods). To guarantee a fair comparison with the SSN and Paired-SSN methods, we used the Pearson Correlation Coefficient (PCC) in LIONESS according to its original instruction [33]. To filter the noise of PCC correlation in the personalized state transition networks, the previously mentioned gene interaction network in Paired-SSN was also used during LIONESS and SSN analysis.

Based on the personalized state transition networks constructed by LIONESS, SSN and Paired-SSN respectively, we applied the NCUA method for identifying the personalized driver genes (Fig 4A). From Fig 4A, we found that considering the CCG and NCG genes as current gold-standard of cancer driver genes, the F-measures of NCUA would be higher when Paired-SSN networks rather than LIONESS and original SSN were used. Therefore, in contrast to other personalized state transition network inferred by other traditional methods such as SSN and LIONESS, Paired-SSN can include more information of individual patients and is suitable for the construction of personalized state transition networks in this work.

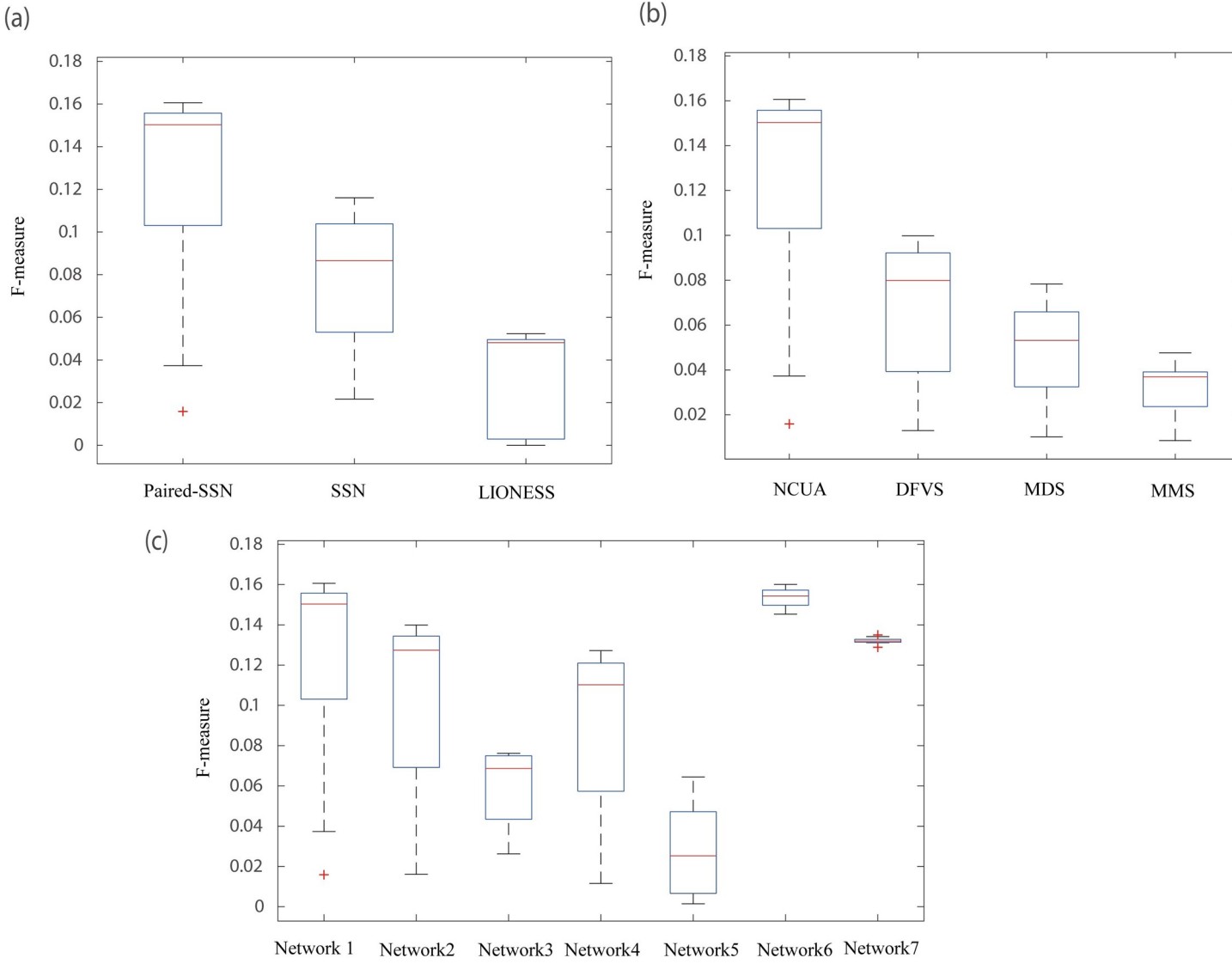

**Fig 4. The performance evaluation of different single sample network construction methods and network control methods.** (a) The significant enrichment F-scores of NCUA on personalized state transition networks constructed by Paired-SSN (the first step of our PNC), SSN and LIONESS. (b)The enrichment F-scores of NCUA (the second step of our PNC) and other structure based network control methods (MMS and MDS and DFVS) on the paired SSN networks (the first step of our PNC) in the list of CCG and NCG genes. (c) The enrichment F-scores of PNC with gene interaction network used in this work (Network 1) and other references (Network 2, Network 3) and the gene interaction network from STRING data set (scores>900) (Network 4) and the gene interaction network with top 10000 high scores from STRING data set (Network 5) and our filleted reliable gene interaction network (Network 6) and our filleted unreliable gene interactions network (Network 7).

## The influence of different network control methods

Furthermore, to evaluate the efficiency of NCUA, we gave computational comparisons in terms of F-measures for identifying personalized driver genes enriched in CCG and NCG genes between NCUA and other methods including MMS (full control)[11], DFVS[18] and MDS[19] (Fig 4B). We should note that for MDS and NCUA, the personalized state transition network is an undirected graph in which the directed information of reference gene interaction is not used. From Fig 4B, we can find that F-measures of NCUA are significantly higher than those of MMS, MDS and DFVS methods on multiple cancer datasets, so that NCUA could efficiently identify personalized driver genes in this study.

## The influence of different reference networks

Furthermore, to estimate the effect of the reference network adopted in PNC, we also used other gene interaction networks as the reference networks, including gene interaction network collected in reference [14] (Network 2), and reference [35] (Network 3) and gene interaction networks from STRING data set (https://string-db.org/) (Network 4). By integrating multiple types of datasets, including Reactome [30], NCI-Nature Curated PID [31] and Kyoto Encyclopedia of Genes and Genomes [32], the Network 2 consists of 5959 genes and 108,281 edges. For the Network 3, 6513 genes and 19,955 synthetic lethal gene pairs were collected the Synthetic Lethality Database (https://omictools.com/synlethdb-tool). By choosing the edge scores higher than 900 from STRING, the Network 4 consists of 12,272 proteins (genes) and 505,116 edges with confidence scores. We also tested the performance of our method on a simulated network with by choosing the top 10, 000 high quality interactions from STRING data set (Network 5). The gene interaction network with 211,794 interactions used in this study is named as Network 1. The performance of PNC with five kinds of gene interaction networks on the 13 cancer datasets were shown in Fig 4C. From Fig 4C, we can see that Network 1 with 211,974 interactions with more complete and higher quality gene interaction information would improve PNC's prediction power.

To further demonstrate the effect of the gene interaction data on PNC, we firstly collected co-expression edges, the literature-derived interactions, and predicted interactions from the 211,794 interactions [28–30, 36], which are noted as unreliable interactions. We removed those unreliable gene interactions and retained 104,153 reliable edges. The reliable interactions (Network 6) and the unreliable interactions (Network 7) are listed in **S5 and S6 files**. We identified the cancer driver genes in 13 cancer data sets by using PNC on Network 1, Network 6 and Network 7 and computed the F-measures for assessing the performance of these three networks (Figs 4C and 5). The results in Figs 4C and 5 can be concluded as,

i. The overall F-measure of Network 6 is higher than that of Network 1 and Network 7 (Fig 4C). Furthermore, the F-measure of Network 6 on COAD, KICH, STAD, THCA, PRAD and HNSC cancer dataset are significantly higher than that of Network 1 while on other cancer datasets, the F-measure of Network 6 is close to that of Network 1 (Fig 5). These results show that removing the unreliable interactions can help to identify more accurate cancer driver genes for some cancer data sets.

ii. The variance of F-measure on Network 1 is rather larger than that of on Network 6 and Network 7. These results show that the integrated gene interaction data would lead to biased predictions, and the filtered reliable network can improve the predictive performance.

The above results show that removing the unreliable interactions can help to identify more accurate cancer driver genes for some cancer data sets. Therefore, a proper reference gene interaction network is a key factor for PNC on identifying cancer driver genes. In the future, we should apply PNC in broad applications by using the reliable interactions among the intensively collected interactions.

## Statistic analysis of the personalized driver genes identified by PNC

In fact, PNC provides a set of personalized driver genes for each patient. For a given cancer dataset, by calculating the frequency of genes that appear as the personalized driver genes among a group of patients, we defined high-frequency personalized driver genes ($fh > 0.6$), medium-frequency personalized driver genes ($0.3 < fm \le 0.6$), and low-frequency personalized

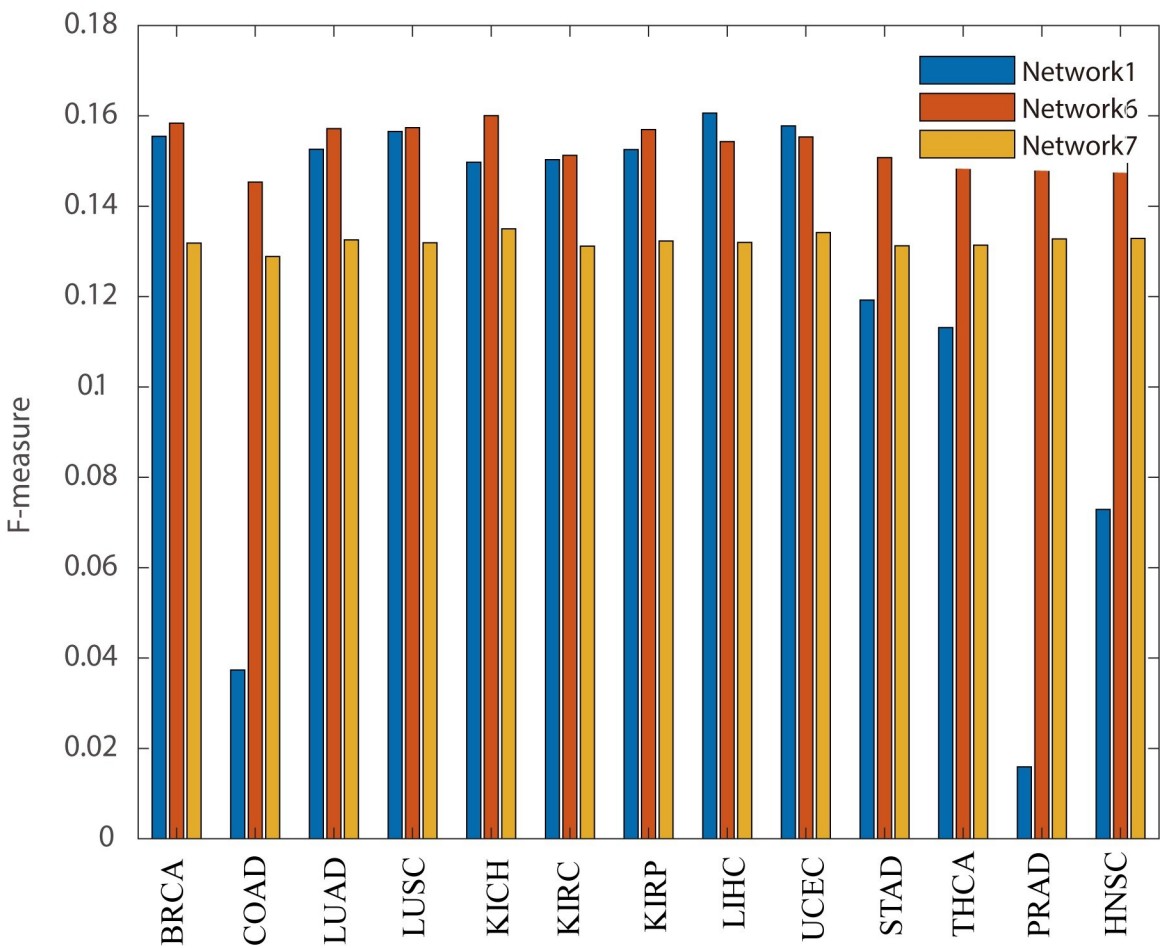

**Fig 5. The F-measures on different cancer data sets by using PNC on gene interaction network used in this work (Network 1) and our filleted reliable gene interaction network (Network 6) and our filleted unreliable gene interactions network (Network 7).**

driver genes (*fl* ≤ 0.3). In Fig 6A, we gave the fractions of high-frequency personalized driver genes (yellow), medium-frequency personalized driver genes (red), and low-frequency personalized driver genes (blue) on 13 cancer datasets. Fig 6A revealed that different cancer datasets had different distributions for personalized driver genes among groups of patients. For example, in the PRAD cancer dataset, the high-frequency personalized driver genes accounted for the majority (>60%); however in the UCEC cancer dataset, the high-frequency personalized driver genes accounted for a minority (<20%). These results demonstrated that the heterogeneity of personalized driver genes varied in different cancer datasets, reflecting the tumor heterogeneity on the level of driver gene profiles.

It is widely accepted that driver genes have high mutation frequency, which is called "long tail of rarely mutated genes" [37]. To demonstrate the ability of discovering the driver genes with rare mutation frequency, we firstly obtained the single nucleotide variations (SNVs) data of 13 kinds of cancer datasets from TCGA. We then divided the personalized driver genes within CCG and NCG genes into driver genes with rare mutation frequency (0<frequency<0.05) and high mutation frequency (frequency>0.05) respectively. Finally we calculated the fraction of personalized driver genes with rare mutation frequency and high mutation frequency respectively on 13 kinds of cancer data as shown in Fig 6B. Results

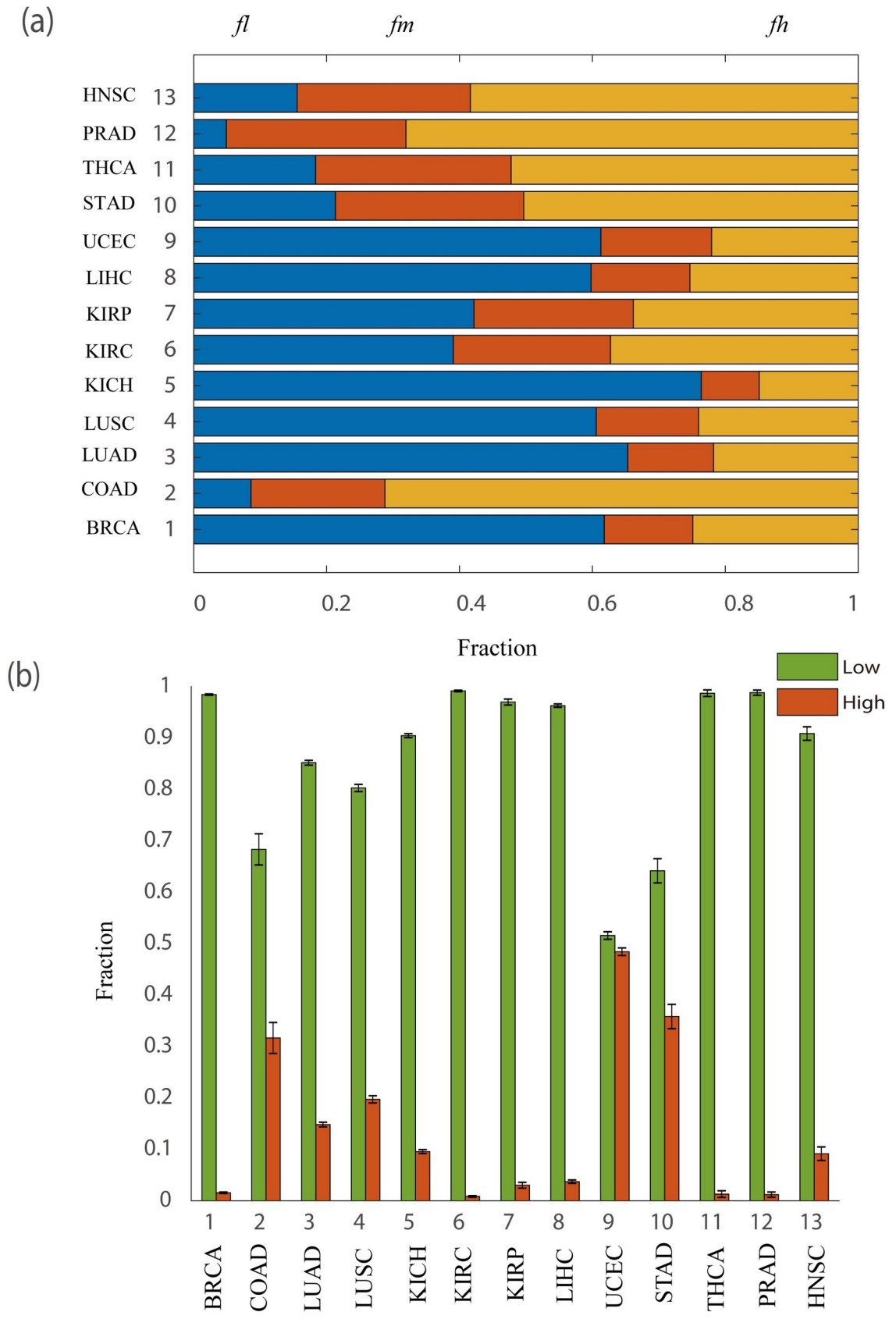

**Fig 6. Statistic analysis of the personalized driver genes.** (a) The fractions of high-frequency personalized driver genes (yellow, *fh*), medium-frequency personalized driver genes (red, *fm*), and low-frequency personalized driver genes (blue, *fl*) on 13 cancer datasets. (b) The fraction of personalized driver genes in CCG and NCG with low mutation frequency and high mutation frequency respectively on 13 kinds of cancer data.

in Fig 6B show that PNC tends to discover personalized driver genes with rare mutation frequency on most of cancer datasets. Of note, the fraction of the driver genes with rare mutation frequency and high mutation frequency are similar on UCEC and STAD cancer datasets.

## The enrichment performance evaluation of the personalized driver genes identified by PNC

To further support the efficiency of PNC by statistic significance, the enrichment p-values of the predicted driver genes in CCG and NCG lists were evaluated for PNC as shown in Fig 7. The enrichment p-values were calculated by using the hyper geometric test [38], representing the significance of the CCG / NCG driver genes rediscovered by PNC. The computational details were shown in section of Methods. From the results of Fig 7, we can see some interesting facts on tumor heterogeneity: i) for the CCG and NCG lists, the enrichment p-values of PNC vary in different cancer datasets. For example, for COAD, PRAD, THCA and HNSC cancer dataset, not all patients have significant enrichment results (Fig 7). ii) But for other cancer datasets such as BRCA, KICH, LUAD and UCEC cancer datasets, the personalized driver genes are significant enriched in CCG and NCG genes (p-value<0.05) for all patients by using the PNC method (Fig 7).

## Novel insights for understanding tumor heterogeneity in cancer by PNC

To illustrate the above demonstrated tumor heterogeneity in various types of cancer, we used the individual specific sub-networks related with driver gene *TP53* and its first-order network neighboring genes as an example. By integrating the sub-networks of patients on each cancer data set, we obtained the statistic information of individual specific sub-networks of *TP53* including frequency as personalized driver genes, the SNVs mutation frequency, mean differential expression fold change (the absolute value of log2 fold change between normal expression data and tumor expression data), and mean network degree in individual specific sub-networks (Table 1). We can see that i) although *TP53* has different mutation frequency in multiple cancer data sets, PNC identified *TP53* having high driver frequency in most of cancer data sets. This result demonstrates that PNC is able to predict individual driver genes for cancer solely based on gene expression without DNA sequence information; ii) *TP53* has low differential expression fold change ($|\log2(\text{fold-change})|<1$) in some cancer data sets (COAD, KIRP, LUSC, STAD, THCA and PRAD) but has high average network degree so as to be hub gene. This result demonstrates that even when the candidate driver genes are hidden in transcription profiles, they can still be explored by their network and structural characteristics, e.g. by PNC.

Next we computed the p-value of the individual-specific *TP53* sub-networks enriching in the KEGG pathways by using the hyper-geometric test (Methods). By integrating the individual-specific pathways in 13 cancer data sets, we obtain the cancer pathways with its patient frequency in multiple cancer data sets. All enriched pathways in 13 cancer data sets are listed in **S7 File**. We found that 44 pathways which appeared in all 13 cancer data sets and 45.45% of them were reported to be related with cancer in previous biological observations. The detail results were listed in **S8 File**. In Fig 8 we showed the patient frequency value of these confident

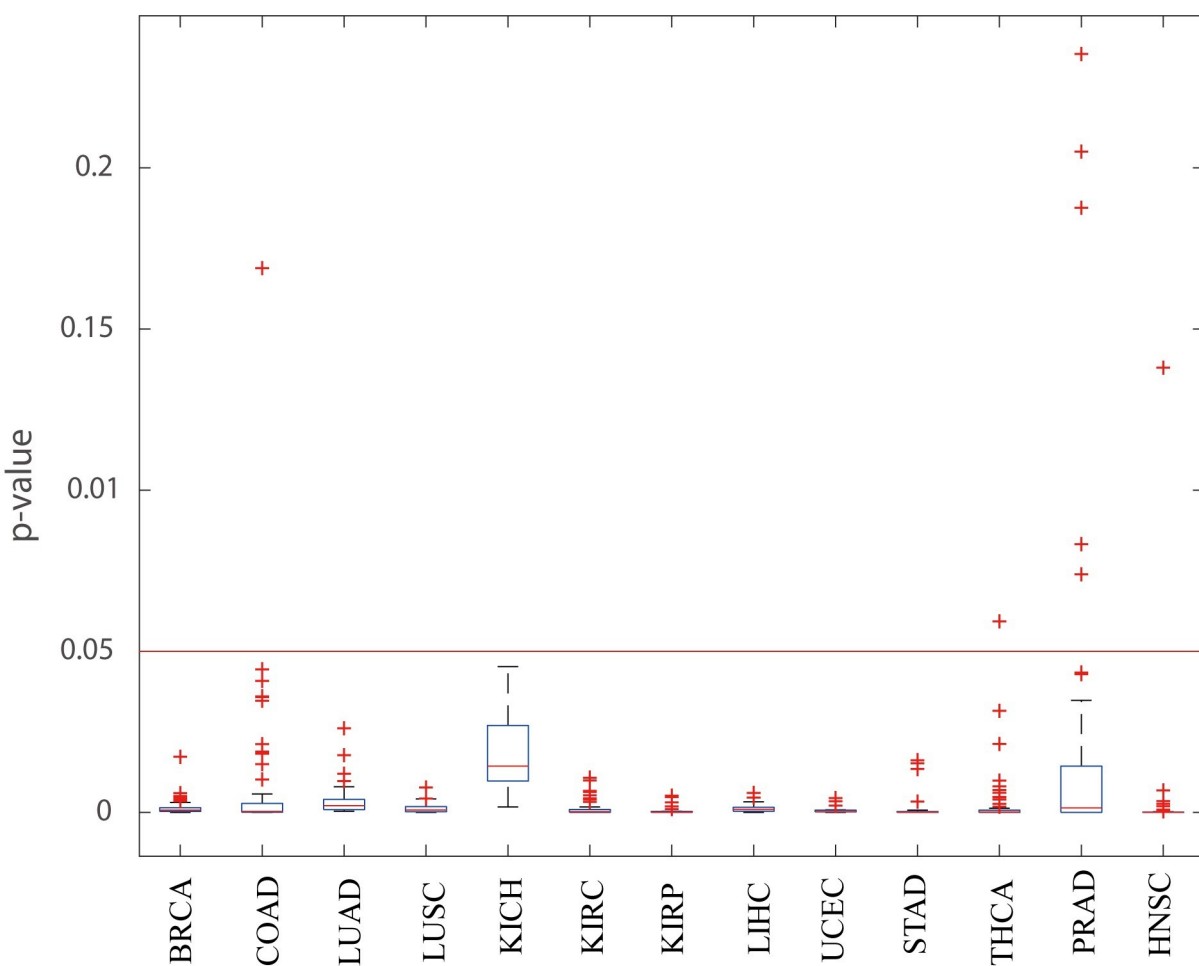

**Fig 7. The p-value of personalized driver genes enriched in CCG and NCG genes on 13 kinds of cancer data.** The red line denotes the significant threshold value 0.05.

**Table 1. The statistic information of individual specific sub-networks of TP53 in cancer.**

| Cancer | Driver frequency | Mutation Frequency | Mean Fold change | Mean degree |
|--------|------------------|--------------------|------------------|-------------|
| BRCA | 1 | 0.3111 | 9.5347 | 380.32 |
| COAD | 0.92 | 0.4870 | 0.2095 | 38.98 |
| LUAD | 1 | 0.4608 | 10.8065 | 387.49 |
| LUSC | 1 | 0.8202 | 5.3138 | 339.75 |
| KICH | 1 | 0.3333 | 23.9574 | 400.52 |
| KIRC | 1 | 0.0160 | 1.0044 | 176.08 |
| KIRP | 1 | 0.0248 | 0.1905 | 243.41 |
| LIHC | 1 | 0.3181 | 8.8303 | 389.20 |
| UCEC | 1 | 0.2862 | 11.3241 | 359.39 |
| STAD | 1 | 0.4775 | 0.0036 | 224.71 |
| THCA | 1 | 0.0074 | 0.3888 | 147.87 |
| PRAD | 0.9423 | 0.0692 | 0.6188 | 147.40 |
| HNSC | 1 | 0.7347 | 1.2537 | 152.72 |

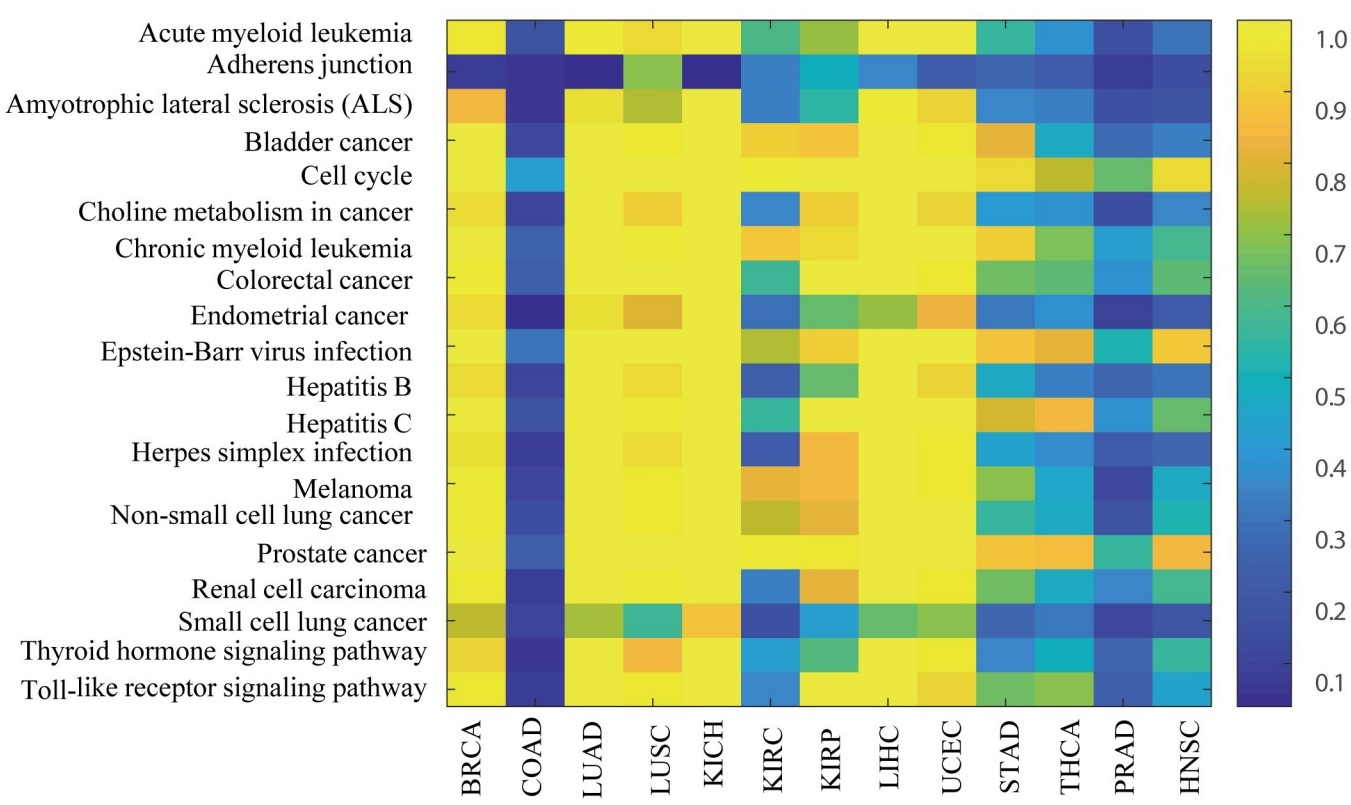

**Fig 8. Heat map of the patient frequency (different colors) of these 20 intersected pathways in 13 cancer data sets which was previous reported in other literatures.**

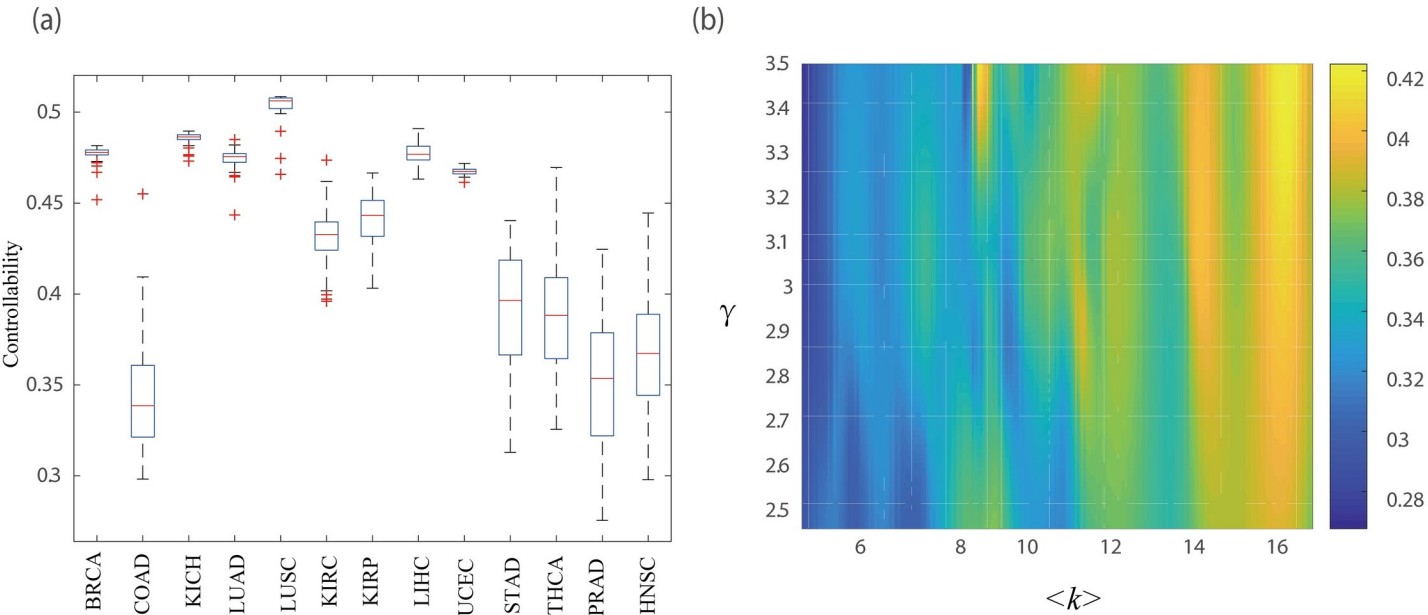

**Fig 9. Controllability evaluation of individual patient system by using PNC.** (a) Box plot with the distribution of the personalized controllability in 13 cancer datasets. (b) Heat map of the personalized controllability (different colors) which is as a function of the average degree and power law degree exponent.

common pathways in 13 cancer data sets. We can see that i) most of these pathways are observed in only a little fraction of the patients in COAD; ii) *cell cycle pathway* is enriched in most of patients in 12 cancer data sets (except COAD), demonstrating that cell cycle genes are regulated by TP53 in most of cancer patients; ii) *Adherens junction pathway* are enriched in most of patients (69.38%) for LUSC cancer data set while it is enriched in small number of patients (<50%) for other cancer data sets. These results demonstrate that there are even different tumor heterogeneity for the same pathway in different cancer types. Therefore, our PNC is capable to reveal some novel insights into understanding the tumor heterogeneity in cancer.

## Controllability evaluation of individual patient system by using PNC

To quantify the personalized controllability of individual patient system, we defined controllability as follows,

$$Controllability = ||D||/||V||, \tag{1}$$

where $||D||$ denotes the number of driver nodes and $||V||$ denotes the number of connected nodes in the network. In Eq 1, the smaller the value of *controllability* is, the more easily the personalized state transition network will be controlled. The personalized controllability of PNC on 13 cancer datasets from TCGA are shown in Fig 9A. We can find that the personalized controllability of individual patients varies in different cancer datasets. For example, the personalized controllability in COAD cancer data is less than 0.4, while the personalized controllability in LUSC is larger than 0.5.

To demonstrate the relationship between personalized controllability and network structure, we firstly obtained the connected component (CC) of each personalized state transition network and then used the CC network to determine the degree exponent γ by using the Kolmogorov-Smirnov goodness-of-fit test [39]. More computational details are shown in Supplementary Note 4 of **S2 File**. The network parameters in each CC on different cancer datasets including number of nodes, average degree and degree exponent are presented in **S9 File**. Finally in Fig 9B we gave the controllability (defined in Eq 1) of personalized state transition networks which are significantly subject to the power-law distribution (i.e., the p-value of the Kolmogorov-Smirnov goodness-of-fit test is greater than 0.05) as a function of the average degree and power law degree exponent. From Fig 9B, we also found that the personalized state transition networks with smaller values of the average degree and power law degree exponent would be easier to control. That is, the more heterogeneous the degree distribution of a personalized state transition network is, the easier it will be to control the entire system.

To more clearly explain how network parameters affect the controllability, we applied NCUA on multiple synthetic scale-free (SF) networks [40], as shown in S1 Fig of **S2 File**. In S1 (a) Fig of **S2 File**, we showed that for γ < 2, the minimum number of input nodes increases as γ increases, while the minimum number of input nodes does not depend on the average degree $<k>$. However, if the value of γ is above 2, the minimum number of input nodes is governed by both γ and $<k>$. Furthermore, SF networks with a large value of γ above 2 or large value of $<k>$ are hard to control, as shown in S1 (a) Fig of **S2 File**. In S2 Fig of **S2 File**, we listed the minimum number of input nodes in the function of the network size for a fixed degree exponent with γ = 1.4, 1.6 and γ = 2.4, 2.6. We found that the minimum number of input nodes decreases with the increasing network size for γ < 2, while for γ > 2, the minimum number of input nodes is not affected by the network size. These results are complemented by S1 (b-c) Fig of **S2 File**, where it shows that, compared with the Erdös-Rényi random networks (ER), only fewer nodes are needed to control the entire network if the power law degree exponent γ

is smaller than 3, whereas it is more difficult to controlled with a value of γ above 3. These results give insight into heterogeneous networks (γ<3) will be easier to control with the minimum number of driver nodes than the homogeneous networks (γ>3). The more heterogeneous an SF network degree distribution is, the easier it is to control the entire system. Those simulation results match well with the results of our analysis results in cancer. More details are shown in Supplementary note 3 and Supplementary Note 5 of **S2 File**.

## Discussion

Cancer is known as a disease mainly caused by gene alterations. Identification of driver genes is a key point of focus in cancer genomics. Computational models and methods are required to prioritize biologically efficient driver genes dependent on cancer high-throughput sequencing data. However, few methods can efficiently distinguish the personalized driver genes that change the state of genes in each patient. Here, we propose PNC from structure based network control perspective to identify personalized driver genes in cancer. PNC considers how to construct the personalized state transition network capturing the phenotype transitions between healthy and disease state through the utilize of Paired-SSN method and find the personalized driver genes on phenotype transitions by applying NCUA, which can control the individual from the normal attractor to the disease attractor.

To further demonstrate the advantage of our PNC, it has been employed to investigate cancer driver genes of cancer patients from TCGA by screening known driver genes in the CCG and NCG genes. We found that in contrast to current state-of-the-art methods, PNC shows a higher performance in identifying the cancer driver genes enriched in the CCG and NCG genes. Especially, PNC revealed that the personalized controllability vary in different cancer datasets and the more heterogeneous the degree distribution of personalized state transition network is, the easier it is to control the entire system. Furthermore, personalized driver genes can be explored by their network characteristics even when they are hidden in transcription and mutation profiles. In addition, we have validated that Paired-SSN has better performance for constructing personalized state transition networks compared with other methods, such as SSN and LIONESS. And NCUA is a better choice for the identification of cancer driver genes compared with other network control methods such as MMS (full control), MDS and DFVS. However, the efforts to achieve promising improvements for the utilization of NCUA approach are still ongoing. For example, NCUA does not consider multiple driver-node sets to control the network. From the perspective of identifying personalized driver genes, the application of NCUA may not be optimal, and this limitation may generate false-positive prediction results in the fields of biology and biomedicine. Therefore in the future it is worth studying how to find multiple driver nodes configurations in large-scale networks, which will benefit the application of structure based network control methods in diverse biology and biomedicine applications. Furthermore NCUA ignores the weight and direction information of network edges, causing a potential bottleneck in characterizing the edge control rather than only node control. Therefore, another important research direction will be the extension of our NCUA method to edge dynamics [41].

## Methods

### Paired-SSN

For the Paired-SSN method, the co-expression network of the tumor sample network or normal sample network for each patient is first constructed based on statistical perturbation analysis of one sample against a group of given reference samples (e.g., choosing the expression data of all normal samples of all patients in a given cancer data set as the reference data here) with

the SSN method [26]. If the differential PCC ($\Delta PCC$) of an edge is statistically significantly large based on the evaluation of the SSN method, this edge would be kept for the single sample (normal sample or tumor sample) from each patient. The p-value for an edge can be obtained from the statistical Z-value measuring $\Delta PCC$. Of note, the $\Delta PCC$ of an edge between genes $i$ and $j$ and its significant Z-score are calculated using the following formula:

$$\Delta PCC_{ij} = |PCC_{ij}^{n+1} - PCC_{ij}^{n}|$$

$$Z_{ij} = \frac{\Delta PCC_{ij}}{(1 - (PCC_{ij}^{n})^2)/(n-1)}, \tag{2}$$

where $PCC_{ij}^{n}$ is the PCC of an edge between genes $i$ and $j$ in the reference network with $n$ samples; and $PCC_{ij}^{n+1}$ is the $PCC$ of the edge between genes $i$ and $j$ in the perturbed network with one additional sample, given that this single sample (e.g. normal sample or tumor sample) for each patient is added to the reference sample group. All of the edges with significantly differential correlations (e.g., p-value < 0.05) are used to constitute the SSN for one normal sample or tumor sample.

To illustrate that the sample-specific network theory can be applied in situations with limited numbers of patients, we generated two series of reference numbers. The length $n$ of the two series (i.e. the number of the reference samples) was chose as 20, 50 and 100 and the correlation of the two series of numbers was a fixed value ($PCCn = 0$). Based on the generated two series of reference numbers, we randomly generated one series of two numbers (gene expression value of two genes for one sample) and obtain the distribution of $\Delta PCCn$. The random simulation was repeated 2,000,000 times, where the value of $\Delta PCCn$ with a significant P-value of 0.05 in the two-tails area was selected from every distribution of simulation (S3 (d) Fig of S2 File). As shown in S3 (a-c) Fig of S2 File, the distribution of $\Delta PCCn$ follows a new type of distribution defined as volcano distribution, whose tail areas are similar to those of a normal distribution in a random condition. At the same time, the significant value of $\Delta PCCn$ with the P-value of 0.05 can also be obtained from Eq 2 (S3 (d) Fig of S2 File). The z-scores of $\Delta PCCn$ with the P-value of 0.05 for the random simulation and the theoretical calculation have little difference (S3 (d) Fig of S2 File). The above results well validates that SSN method can be suitable for situation with limited numbers of patients.

Then, the personalized differential co-expression network between the normal sample network and tumor sample network can be constructed in which the edge between gene $i$ and gene $j$ will exist if the p-value of the edge is less than (greater than) 0.05 in the tumor network but greater than (less than) 0.05 in the normal network for their corresponding patient. To more precisely demonstrate the regulatory mechanism of personalized state transition networks, we use the gene interaction network to filter the noise of PCC between gene pairs. Finally the personalized state transition networks are obtained in which edges exist in both the gene interaction network and the differential co-expression network for each patient.

## LIONESS

LIONESS reconstructs the individual specific network in a population of tumor samples as the personalized state transition network for each tumor sample [33]. LIONESS constructs the state transition network by calculating the edge statistical significance between all of the tumor samples and the tumor samples without a given single sample. In order to guarantee a fair comparison with SSN and Paired-SSN, we use PCC in LIONESS to model the sample-specific state transition network. The network specific to one sample in terms of the aggregate

networks is then:

$$e_{ij}^q = N_s(e_{ij}^{N_s} - e_{ij}^{N_s-q}) + e_{ij}^{N_s-q},$$ (3)

where $e_{ij}^{N_s}$ and $e_{ij}^{N_s-q}$ are the PCC values of gene pairs $i$ and $j$ in sample $n$ and all but sample $q$, respectively; $N_s$ is the number of the samples. To construct the sample-specific network, after we obtain the PCC absolute value distribution $S$ of all gene pairs we choose a threshold to determine the differential expression edges in the sample-specific network as follows:

$$w = \mu(S) + 2\delta(S),$$ (4)

where $\mu(S)$ and $\delta(S)$ are the mean and standard variance of the PCC absolute value distribution $S$ of all gene pairs.

## NCUA

In many biological systems, there is adequate knowledge of the underlying wiring diagram, but not of the specific functional forms [42, 43]. The real complex systems especially the biological systems usually have hundreds and thousands of nodes so that the classical control theory are prohibited [11–13]. From the dynamics point of view, it is transcriptional networks that thus determine the state transition of gene expressions of an individual patient at different time points in cancer development. To be specific, human fibroblasts could be reprogrammed into an induced pluripotent stem cells (iPSCs), via adding four transcription factor POU5F1, SOX2, KLF4, and MYC to fibroblasts cells [44]. Indeed, transcriptional networks are composed of numerous nodes linked via a complex set of interactions, and the dynamical behavior are controlled by few number of driver nodes. To understand the system dynamical behavior, it is necessary to investigate the relationship between the structure and the dynamics of complex systems. Recently, structure-based network control approaches, which only should know whether there are edges or not in the networks, connects the structure with the dynamics of complex systems and gives us the ability to steer the states of complex systems to the desired states through a minimum set of driver nodes [11]. The structure-based network control approaches are useful for complex biological networks because we usually know structures of them but do not know the specific interaction strengths. In past decades, the focus is shifting to the methods for studying the structure-based network control of systems with nonlinear dynamics [45–47]. Recently, the method of Feedback Vertex Set (FVS)-based control (FC) method [16, 17] has been successfully applied to large scale complex biological networks with known network topologies but without knowing the nonlinear dynamical equations that govern their state transition. It only requires the functional form of the governing equations to satisfy some continuous, dissipative, and decaying properties. To drive the state of a network to any one of its naturally occurring end states (i.e., dynamical attractors), FC needs to manipulate a set of nodes (i.e., the FVS) that intersects every feedback loop in the network. Under the FC framework, DFVS is proposed to study the structural control problem of directed networks, by selecting a minimum set of driver nodes to render a directed network structural controllable, which has been recently discussed [18].

DFVS assumes that the state of the nodes in a directed network, characterized by source nodes $s_j(t)$ and internal nodes $x_i(t)$, obeys the following equations:

$$dx_i/dt = F_i(x_i, I_i, t),$$
$$ds_j/dt = G_j(t)$$ (5)

where $I_i$ is a set of predecessor nodes of node $i$, $F_i$ captures the nonlinear response of node $i$ to its predecessor nodes $I_i$, and $G_j$ governs the nonlinear response of source node $s_i$.

If the undirected networks are considered as bi-directional networks, there are no source nodes in the undirected network nodes. Therefore Eq 5 of an undirected network $G\ (V, E)$, is reduced to,

$$dx_i/dt = F_i(x_i, I_i), \tag{6}$$

where $x_i$ denotes the state variable of the $i$-th node at time $t$ in undirected networks, $I_i$ is a set of neighborhood nodes of node $i$, and $F_i$ captures the nonlinear response of node $i$ to its neighborhood nodes in undirected networks.

In fact, the selection of driver nodes may be dependent on the network style (directed or undirected). Motivated by this fact, under the FC framework we formalized the Nonlinear structural Control problem of the Undirected networks (NCU), i.e., how to choose the minimum set of driver nodes for changing the network state from one attractor to the desired attractor in large-scale networks with symmetric edges information. More details can be found in Supplementary Note 1 of **S2 File**. Here we develop a graphic-theoretic algorithm called Nonlinear Control of Undirected network Algorithm (NCUA) under the nonlinear FC framework for determining the minimum driver nodes in undirected networks. Since NCUA and DFVS study the structural network control of undirected and directed biological networks respectively based on the framework of FC, therefore NCUA and DFVS methods completely depict the structure based network control of the large scale system with nonlinear dynamics from the respective of FC.

By assuming that each bi-directional edge is considered as a feedback loop, our NCUA searches the sets of minimum nodes whose removal leaves the graph without edges in undirected networks (Fig 1). The implementation of NCUA consists of two main steps: (i) constructing a bipartite graph from the original undirected network, in which the nodes of the top side are the nodes of the original graph and the nodes of the bottom side are the edges of the original graph, and (ii) determining the MDS of the top side nodes to cover the bottom side nodes in the bipartite graph by using Integer Linear Programming (ILP). The computational details are as follows,

The NCUA firstly construct a bipartite graph from the original undirected network. For a given undirected network $G\ (V, E)$, we assume that each edge is bi-direct, converting $G\ (V, E)$ into a bipartite graph $G\ (V_T, V_\perp, E_1)$, where $V_T \equiv V$ and $V_\perp \equiv E$. If $v_i \in V_T$ was one of the nodes for $v_j \in V_\perp$, we added one edge connecting $v_i \in V_T$ and $v_j \in V_\perp$ into the set $E_1$. After obtaining the bipartite graph $G\ (V_T, V_\perp, E_1)$, NCUA selects a minimum dominating set $S$ from $V_T$, which must cover all the nodes in $V_\perp$. This minimum dominating set is used to determine the driver nodes for controlling the whole network. In NCUA, we select the driver nodes by solving the following ILP:

$$\min f = \sum_{v \in V_T} y_v$$
$$s.t. \sum_{\{v,u\} \in E_1} y_v \geq 1 (\text{every} u \in V_\perp), y_v \in \{0, 1\} \tag{7}$$

where $\sum_{v \in V} y_v$ denotes the number of candidate nodes and $y_v$ is an indicative variable; $\{v, u\} \in$ $E1$ denotes the edge connecting $v_v$ and $v_u$ in the bipartite graph $G\ (V_T, V_\perp, E_1)$; when node $v$ in the up side node set in the bipartite network is within the driver nodes, $y_v = 1$, and otherwise, $y_v = 0$;

As is known, problem (7) is an NP-hard (non-deterministic polynomial-time) hard) problem [48]. The NP-hard problem, in computational complexity theory is defined as: *a problem H is NP-hard when every problem L in NP can be reduced) in polynomial time to H; that is, assuming a solution for H takes 1 unit time, we can use H's solution to solve L in polynomial time* [48]. As a consequence, finding a polynomial algorithm to solve any NP-hard problem would give polynomial algorithms for all the problems in NP, which is unlikely as many of them are considered difficult[48]. Although it is an NP-hard problem, the optional solution can be obtained for the moderate size graphs with up to a few tens of thousands of nodes by utilizing the LP-based classic branch and bound method [49].

To easier understand NCUA, we gave the concept comparisons including the network types and targeted states and time complexity and network dynamics between our NCUA and other network control methods including MMS [11–13], DFVS[18] and MDS[19] (S2 Table of **S2 File**). In S4 Fig of **S2 File**, we used an example to intuitively explain the difference between the NCUA and MMS, MDS and DFVS. We summarized some key points of different structure based network control methods as follow:

i. The MMS control methods investigate the structural controllability of directed networks with linear or local nonlinear dynamics through a minimum set of input nodes and they only give an incomplete view of the network control properties for a system with nonlinear dynamics.

ii. MDS control method studies the structural controllability of undirected networks by assuming that each driver node in the MDS model can control its associated edges independently in the undirected networks. Since MDS works with the strong assumption that the controllers can control its outgoing links independently, it requires higher costs in many kinds of networks which may underestimate the structural control analysis of undirected networks.

iii. NCUA and DFVS study the structural controllability of undirected and direct networks respectively based on the framework of FC. Therefore NCUA and DFVS methods ultimately depict the structural controllability of the large-scale system with nonlinear dynamics from the respective of FC. Since the FC control method [16, 17] assumes that the functional form of the governing equations must satisfy some continuous, dissipative, and decaying properties, DFVS and NCUA may be only suitable some specialized nonlinear systems.

### Hub genes selection method

The Hub genes selection method regards the hub genes in the constructed network as driver genes. After obtaining the degree distribution of all genes $U$ in the personalized state transition network, we use a threshold as introduced in following formula to obtain the hub genes,

$$w = \mu(U) + 2\delta(U), \tag{8}$$

where $\mu(U)$ and $\delta(U)$ are the mean and standard variance of the degree distribution $U$ of all gene, respectively.

### F-measure

To verify the effectiveness of the PNC analysis mainly based on structure based network control methods, the F-measure is considered to assess the enrichment ability of identified personalized driver genes in a given gold standard list (CCG and NCG genes), which considers both

the precision) and the recall) using the formula:

$$F_i = \frac{2P_i R_i}{P_i + R_i},$$

(9)

where $P_i$ denotes the fraction of correctly identified genes among all of the identified genes (precision) and $R_i$ denotes the fraction) of correctly identified personalized driver genes among the given dataset (recall).

## Enrichment analysis of the personalized driver genes in gold-standard cancer driver gene lists

To estimate the significance of overlap between the predicted personalized driver genes for PNC method and a gold-standard cancer driver gene lists such as Cancer Census Genes and Network of Cancer Genes, we compute the $p$-value by the hyper geometric test [38] as follow,

$$p = P(X \geq k) = \sum_{k=1}^{\infty} \frac{\binom{K}{k}\binom{N-K}{n-k}}{\binom{N}{n}},$$

(10)

where $N$ is the number of genes in background gene interaction network, $K$ is the number of gold-standard cancer driver gene lists, $k$ is the number of the predicted personalized driver genes overlapping with the genes in the gold-standard cancer driver gene lists, and $n$ is number of the predicted personalized driver genes. If the enrichment $p$-value is less than 0.05, then we regard that this predicted personalized driver genes for PNC method is significantly enriched in the gold-standard cancer driver gene lists.

## Enrichment analysis of the individual-specific sub-networks in KEGG pathways

We computed the p-value of the individual specific sub-networks of driver gene (driver gene and its first-order network neighboring genes) enriching in the Kyoto Encyclopedia of Genes and Genomes (KEGG) pathways by using formula (10), where $N$ is the number of all pathways genes, $K$ is the number of a given pathway genes, $n$ is number of genes in the individual-specific sub-network, and $k$ is the number of genes in the individual-specific sub-network overlapping with the genes in a given pathway. If the calculated p-value was less than 0.05, then we regarded that this individual-specific sub-network is significantly enriched in a given KEGG pathway.

## Supporting information

**S1 File. The full list of integrated networks with 211,794 interactions.**
(XLSX)

**S2 File. Supplementary manuscript of a novel network control model for identifying personalized driver genes in cancer.**
(DOCX)

**S3 File. The list of Cancer Census Genes (CCG) and Network of Cancer Genes (NCG) to assess the F-measure of the identified personalized driver genes of different methods.**
(XLSX)

**S4 File. The predicted driver genes list of 14 different methods.**
(XLSX)

**S5 File. The list of the reliable interactions after removing co-expression edges, the literature-derived interactions and predicted interactions from the full interactions described in S1 File.**
(XLSX)

**S6 File. The list of the unreliable interactions such as co-expression edges, the literature-derived interactions and predicted interactions.**
(XLSX)

**S7 File. The results of enriched pathways with its patient frequency in multiple cancer data sets.**
(XLSX)

**S8 File. The 44 intersected pathways appearing in all 13 cancer data sets.**
(XLSX)

**S9 File. The statistic results of personalized network using the Kolmogorov-Smirnov goodness-of-fit test on 13 kinds of cancer datasets from TCGA.**
(XLSX)

## Acknowledgments

We thank Professor Liu Xiaoping from Shandong University, China and Professor Tatsuya Akutsu in Kyoto University, Japan for giving us assistances and valuable comments during the preparation of manuscript.

## Author Contributions

**Conceptualization:** Wei-Feng Guo, Tao Zeng, Luonan Chen.

**Data curation:** Wei-Feng Guo, Yan Li.

**Funding acquisition:** Shao-Wu Zhang, Luonan Chen.

**Methodology:** Wei-Feng Guo, Tao Zeng, Jianxi Gao.

**Software:** Wei-Feng Guo.

**Supervision:** Shao-Wu Zhang, Jianxi Gao.

**Visualization:** Yan Li.

**Writing – original draft:** Wei-Feng Guo.

**Writing – review & editing:** Wei-Feng Guo, Shao-Wu Zhang, Tao Zeng, Jianxi Gao, Luonan Chen.

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
