## [Decision Letter · Decision Letter 0]

30 Jul 2019

Dear Dr Chen,

Thank you very much for submitting your manuscript 'A novel network control model for identifying personalized driver genes in cancer' for review by PLOS Computational Biology. Your manuscript has been fully evaluated by the PLOS Computational Biology editorial team and in this case also by independent peer reviewers. The reviewers appreciated the attention to an important problem, but raised some substantial concerns about the manuscript as it currently stands. While your manuscript cannot be accepted in its present form, we are willing to consider a resubmission in which the issues raised by the reviewers have been adequately addressed.

Sincerely,

Feixiong Cheng, Ph.D.

Guest Editor

PLOS Computational Biology

Ruth Nussinov

Editor-in-Chief

PLOS Computational Biology

[LINK]

Reviewer's Responses to Questions

**Comments to the Authors:**

Reviewer #1: This manuscript proposes a novel network control-based method for personalized cancer driver gene identification. This method is composed of two parts: a paired single sample network construction method to construct the personalized state transition; and a structure network control method to identify personalized driver genes. The influence of each part of the proposed method was examined. The results from the proposed method were also analyzed in terms of frequencies of personalized driver genes in different cancer types and proportion of genes with rare mutations detected in all the driver genes.

This paper lacks an in-depth and comprehensive introduction to other methods. See (https://academic.oup.com/bib/article/17/4/642/2240387) figure 1 and (https://academic.oup.com/nar/article/47/8/e45/5324448) figure 1 for a list of methods.

A major problem for this research is the performance comparison. For the comparison, only a few methods were compared (DEG-based, hub-based), whereas none of the more advanced methods (e.g., some methods in the aforementioned two figures) were compared (except for SCS, which was compared in only one dataset). Here are some methods to consider for comparison:

https://academic.oup.com/nar/article/47/8/e45/5324448

https://www.biorxiv.org/content/10.1101/456723v3

https://www.ncbi.nlm.nih.gov/pubmed/24516372

https://www.ncbi.nlm.nih.gov/pmc/articles/PMC4148527/

As the performance is a very important aspect of a new method, only when through comparison to current state-of-the-art methods are conducted can one draw conclusion about the value of the new method.

Please check the GitHub repo. The code can't be downloaded.

Reviewer #2: This work proposed the personalized network control model (PNC) to identify the personalized driver genes. The authors firstly designed a paired single sample network construction method (Paired-SSN) to construct personalized state transition network. Then a novel structure network control method (NCUA) was presented to identify personalized driver genes. The model is interesting, but the following concerns should be addressed.

1. The authors mentioned that the existing methods mainly focus on cohort-level driver gene identification, and only compared their model with some simple methods in the manuscript. However, several methods aimed to identify personalized driver genes, such as DawnRank (PMID: 25177370), OncoIMPACT (PMID: 25572314), even SSN (PMID: 27596597), have been proposed. The authors should compare their model with these methods.

2. The authors should give more details about how to construct co-expression network for each patient.

3. The driver genes identified by this model were personalized. However, the existing methods they compared were aimed to identify cohort-level driver genes. The authors should describe how to summarize the personalized driver genes at the cohort-level, such as intersection, union, or some other methods.

4. In P16, the authors said “in the KIRP cancer dataset, NCUA on Paired-SSN had higher F-measure than SSN method......”. But it was lower in Figure 3(a).

5. The authors should give the full name or brief description about LIONESS when it was firstly appeared in P11.

Reviewer #3: General comments:

This paper describes a novel method that utilizes a personalized network control model (PNC) and a structure network control method to identify the personalized driver genes by integrating the gene expression data of individual cancer patients and gene interaction network. Some components of the methodology are unnecessarily complicated and not justified well. Some datasets used in the study could be chosen more wisely. Additionally, more details are needed in order to completely evaluate the significance of the results, and a more high-level description of the proposed method would help readers to understand the intuition behind the method. The text of this paper is too theoretical and mathematical and lacks corresponding biological interpretation. It may provide valuable insight if the following concerns are addressed.

Major Comments:

a.In the last paragraph of introduction part, the authors stated that “our PNC model provides a new powerful tool for identifying the personalized driver genes of individual patients and gives novel insights for understanding tumor heterogeneity in cancer.”, However, how the PNC gives novel insights for understanding tumor heterogeneity in cancer should be illustrated in the paper with at least some case studies, which is currently missing.

b.The biological intuition behind the PNC model is missing. It would be interesting if the authors can provide some examples of how their PNC model builds the personalized state transition network, what the biological meaning of the state transition network is. This could provide some biological interpretation of the PNC model.

c.Some high-level description of how the structure network control method takes advantage of the Feedback Vertex Set (FVS)-based control (FC) theory to identify the driver genes and how does it differ from related methods can help readers understand the true intellectual contribution of the paper without figuring out all the mathematical details.

d.The gene interaction network in this paper contains 11,648 genes and 211,974 interactions. There are too many edges, are there any predicated protein-protein interactions included? Are the authors used the protein-protein interactions from the STRING database? In figure1, in the step1 of paired-SSN, the toy figure of the gene interaction network is generated from the STRING database. Are those 211,974 interactions all validated by at least in-vitro experiments in the database? With 211,974 interactions, I would assume a lot of false-positive edges are included, thus leads to biased predictions. If the gene interaction network only has 10,000 very high-quality interactions, will the PNC model still work?

e.The Cancer Census Genes (CCG) and Network of Cancer Genes (NCG) are used as the gold-standard of cancer driver genes, however, are the CCG and NCG are specific for individual patients or not? If the CCG and NCG are not patient-specific driver genes, why should the authors use them as ground-truth? If the CCG and NCG are patient-specific driver genes, there is no need to use such a complex algorithm to predict the personalized driver genes.

f.The authors compared the PNC with DEG-Fold and Hub gene selection, did they compare with somatic mutations, gene amplification, and copy-number changes?

g.In the supplementary note 2, the number of paired samples is less than 80 in 12 cancer types, with such litter number, a lot of theoretical assumptions are not true due to n is small, even n=200 is very different from the assumption . The assumption regarding the distribution of PCC, the distribution of ∆PCC and so on are not true. The sample-specific network theory can’t be applied in such a situation with limited numbers of patients.

h.I am not sure how the driver genes are defined?

i.There is no time-serious data, I am not sure how many states does the nodes have in the personalized state transition networks, how to determine the transition states? How to characterize the state transition of each individual in the PNC? Also in the supplementary file, in equation s2, t→∞, I am not sure what the t represents physically? As all the data are static.

j.In Table S2, the NCUA is an NP-hard problem, i.e., every time one performs the NUCA, and the result can be different.

**Have all data underlying the figures and results presented in the manuscript been provided?**

Reviewer #1: Yes

Reviewer #2: None

Reviewer #3: Yes

PLOS authors have the option to publish the peer review history of their article (what does this mean?). If published, this will include your full peer review and any attached files.

Reviewer #1: No

Reviewer #2: No

Reviewer #3: No

---

## [Decision Letter · Decision Letter 1]

3 Oct 2019

Dear Dr Guo,

Thank you very much for submitting your manuscript, 'A novel network control model for identifying personalized driver genes in cancer', to PLOS Computational Biology. As with all papers submitted to the journal, yours was fully evaluated by the PLOS Computational Biology editorial team, and in this case, by independent peer reviewers. The reviewers appreciated the attention to an important topic but identified some aspects of the manuscript that should be improved.

We would therefore like to ask you to modify the manuscript according to the review recommendations before we can consider your manuscript for acceptance. Your revisions should address the specific points made by each reviewer and we encourage you to respond to particular issues Please note while forming your response, if your article is accepted, you may have the opportunity to make the peer review history publicly available. The record will include editor decision letters (with reviews) and your responses to reviewer comments. If eligible, we will contact you to opt in or out.raised.

- Supporting Information uploaded as separate files, titled 'Dataset', 'Figure', 'Table', 'Text', 'Protocol', 'Audio', or 'Video'.

We hope to receive your revised manuscript within the next 30 days. If you anticipate any delay in its return, we ask that you let us know the expected resubmission date by email at ploscompbiol@plos.org.

Sincerely,

Feixiong Cheng, Ph.D.

Guest Editor

PLOS Computational Biology

Ruth Nussinov

Editor-in-Chief

PLOS Computational Biology

[LINK]

Dear Dr. Guo,

Thank you for submitting the revised manuscript. Two reviewers are satisfied your revision, however, one reviewer pointed out several key issues as well. For example, quality of networks used in you study have to be balanced and discussed carefully. Please respond those comments carefully.

Reviewer's Responses to Questions

**Comments to the Authors:**

Reviewer #1: The authors have addressed my concerns.

Reviewer #2: The authors have addressed all my concerns.

Reviewer #3: General comments:

The authors have addressed most of my questions, however, based on the responses, I have the following comments:

a. The authors have used a gene interaction network from multiple types of datasets, including the gene co-expressions and text-mined interactions. However, these two types of gene interactions are unreliable and with low quality. The authors might need to remove those gene interactions, and re-do the corresponding analysis. As the edges of the integrated networks are still too much, 211,974 interactions, removing the unreliable interactions might help to reduce the number of edges.

b. According to Figure 3, the variance of the proposed PNC approach is rather larger than other approaches, actually, the performance of PNC is comparable to the frequency approach. Also, a lot of previously published approaches, such as DriverNet, OncoDriveFM, and SCS, their performances are very poor and might not as good as even the frequency approach. I guess that using the gene interaction data from the DawnRank approach might lead to biased predictions and comparisons.

**Have all data underlying the figures and results presented in the manuscript been provided?**

Reviewer #1: None

Reviewer #2: Yes

Reviewer #3: Yes

PLOS authors have the option to publish the peer review history of their article (what does this mean?). If published, this will include your full peer review and any attached files.

Reviewer #1: No

Reviewer #2: No

Reviewer #3: No

---

## [Editor Report · Decision Letter 2]

30 Oct 2019

Dear Dr Guo,

We are pleased to inform you that your manuscript 'A novel network control model for identifying personalized driver genes in cancer' has been provisionally accepted for publication in PLOS Computational Biology.

In the meantime, please log into Editorial Manager at https://www.editorialmanager.com/pcompbiol/, click the "Update My Information" link at the top of the page, and update your user information to ensure an efficient production and billing process.

One of the goals of PLOS is to make science accessible to educators and the public. PLOS staff issue occasional press releases and make early versions of PLOS Computational Biology articles available to science writers and journalists. PLOS staff also collaborate with Communication and Public Information Offices and would be happy to work with the relevant people at your institution or funding agency. If your institution or funding agency is interested in promoting your findings, please ask them to coordinate their releases with PLOS (contact ploscompbiol@plos.org).

Thank you again for supporting Open Access publishing. We look forward to publishing your paper in PLOS Computational Biology.

Sincerely,

Feixiong Cheng, Ph.D.

Guest Editor

PLOS Computational Biology

Ruth Nussinov

Editor-in-Chief

PLOS Computational Biology

---

## [Editor Report · Acceptance letter]

18 Nov 2019

PCOMPBIOL-D-19-01026R2 

A novel network control model for identifying personalized driver genes in cancer

Dear Dr Guo,

I am pleased to inform you that your manuscript has been formally accepted for publication in PLOS Computational Biology. Your manuscript is now with our production department and you will be notified of the publication date in due course.

With kind regards,

Matt Lyles
